# Environmental Status of *Cryptococcus neoformans* and *Cryptococcus gattii* in Colombia

**DOI:** 10.3390/jof7060410

**Published:** 2021-05-24

**Authors:** Briggith-Nathalia Serna-Espinosa, Diomedes Guzmán-Sanabria, Maribel Forero-Castro, Patricia Escandón, Zilpa Adriana Sánchez-Quitian

**Affiliations:** 1Grupo de Investigación Ciencias Biomédicas, Escuela de Ciencias Biológicas, Facultad de Ciencias, Universidad Pedagógica y Tecnológica de Colombia, Avenida Central del Norte 39-115, Tunja 150003, Colombia; briggith.serna@uptc.edu.co (B.-N.S.-E.); maribel.forero@uptc.edu.co (M.F.-C.); 2Grupo de Investigación Gestión Ambiental, Departamento de Biología y Microbiología, Facultad de Ciencias e Ingeniería, Universidad de Boyacá, Carrera 2a Este No. 64-169, Tunja 150003, Colombia; dguzman@uniboyaca.edu.co; 3Grupo de Microbiología, Instituto Nacional de Salud, Calle 26 No. 51-20, Bogotá 111321, Colombia; pescandon@ins.gov.co

**Keywords:** *Cryptococcus neoformans*, *Cryptococcus gattii*, environment, Colombia, ecology

## Abstract

The genus *Cryptococcus* comprises more than 80 species, including *C. neoformans* and *C. gattii*, which are pathogenic to humans, mainly affecting the central nervous system. The two species differ in geographic distribution and environmental niche. *C. neoformans* has a worldwide distribution and is often isolated from bird droppings. On the contrary, *C. gattii* is reported in tropical and subtropical regions and is associated with *Eucalyptus* species. This review aims to describe the distribution of environmental isolates of the *Cryptococcus neoformans* species complex and the *Cryptococcus gattii* species complex in Colombia. A systematic investigation was carried out using different databases, excluding studies of clinical isolates reported in the country. The complex of the species of *C. gattii* is recovered mainly from trees of the genus *Eucalyptus* spp., while the complex of the species of *C. neoformans* is recovered mainly from avian excrement, primarily *Columba livia* (pigeons) excrement. In addition, greater positivity was found at high levels of relative humidity. Likewise, an association was observed between the presence of the fungus in places with little insolation and cold or temperate temperatures compared to regions with high temperatures.

## 1. Introduction

The genus *Cryptococcus* includes yeast fungi that develop a specific biotrophic association with different host plants and comprises more than 80 species, with two being pathogenic to humans. Previously known as a single species, *Cryptococcus neoformans* was divided into two varieties, *C. neoformans* variety *gattii and C. neoformans* variety *neoformans* [1,2]; nowadays, these two varieties are classified as two different anamorphic species, *C. neoformans* species complex and *C. gattii* species complex [2,3,4,5]. *C. neoformans* complex has several varieties (*C. neoformans* variety *grubii* serotype A, with three genotypes VNI, VNB, VNII; and *C. neoformans* var. *neoformans* serotype D with genotype VNIV; the serotype hybrid AD is associated with the VNIII genotype), while the complex of the species of *C. gattii* is divided into four genotypes (VGI, VGII, VGIII, and VGIV) [6,7,8,9]. VGII has been subclassified into three associated genotypes (VGIIa, VGIIb, and VGIIc) [10,11]. Nevertheless, it has been suggested that the actual classifications of *C. neoformans* species (known as *C. neoformans* variety *grubii* serotype A with VNB genotype, and VNI / VNII genotypes), *C. deneoformans* (referred to as *C. neoformans* var. *neoformans* with serotype D y genotype VNIV), and a hybrid classification composed of *C. neoformans* and *C. deneoformans* (with serotype AD and genotype VNIII) be recognized. The complex of the species of *C. gattii* has been reorganized as five species: *C. gattii* with genotype VGI, *C. deuterogattii* with genotype VGII, *C. bacillisporus* with genotype VGIII, *C. tetragattii* with genotype VGIV, and finally *C. decagattii* with genotypes VGIV and VGIIIc, in addition to hybrids such as *C. deneoformans* with *C. gattii*, the hybrid *C. neoformans* with *C. gattii,* and the hybrid *C. neoformans* with *C. deuterogattii* [6,7]. These species can cause pulmonary effects and damage to the central nervous system (CNS) [8], with being *C. neoformans* the worldwide cause of cryptococcal meningoencephalitis [9]. The *C. neoformans* complex affects immunocompromised individuals, mainly those affected by the human immunodeficiency virus (HIV) [12,13,14,15]. In contrast, the *C. gattii* complex is associated with infections in apparently immunocompetent patients, indicating that even people with a healthy immune system can be affected by this pathogen [16,17]. It should be noted that *Cryptococcus* spp. is capable of remaining latent without affecting the host, which makes it highly adaptive [9].

The *C. neoformans* complex has a wide distribution in different departments of Colombia, such as Cauca [18], Córdoba [19], Cundinamarca [20,21], Huila [22], Nariño [23] Norte de Santander [24,25,26,27], and Valle del Cauca [28,29]. In the same way, other regions of the world, such as Malawi [30], Nigeria [31,32], China [33], South Africa [34], Brazil [35,36], United States [37], Italy [38], Guinea [39], Argentina [40,41], and Ecuador [42], among others [25,43,44,45,46,47], have reported this species in the environment. This complex is generally associated with birds’ excreta, especially from pigeons (*C. livia*) [48], saprophytic sources of the *C. neoformans* complex and the soil contaminated with it [49,50,51,52]. The high contents of nitrogen, creatinine, and salts present in these wastes generate an appropriate environment for the growth of the fungus [53,54]. The pigeon feces, having low moisture content and low exposure to sunlight, are reservoirs for the *C. neoformans* complex [23]. Additionally, the *C. neoformans* complex has been reported in tree species, with higher prevalence compared to the *C. gattii* complex, for example in *Eucalyptus* spp. [55,56], *Terminalia catappa* [22], *Olea europea* [57], and *Ceratonia* (carob tree) [55].

Conversely, the *C. gattii* complex has been described in temperate regions such as Canada, California, and Oregon [58]. Thus, in Colombia, this species has been identified in the regions of Norte de Santander [25,26], Cundinamarca [59,60], and Córdoba [19], and across the globe, including Australia [61], Africa [34], India [45,62], Italy [55], the United States [63] and southern California [64], Canada [58], Spain [65], and China [33], among other regions [45,66,67,68,69,70,71,72,73].

The habitats of *C. gattii* complex are associated with tree species, with reported isolation from flowers, bark, and leaves [46,74,75]. Despite this, this complex has been found in the environment to a lesser degree than the *C. neoformans* complex, especially when associated with the genus *Eucalyptus* [55]. However, at present, it has also been associated with flowers, hollows, and decomposing wood of different trees, including *Acacia*, *Ficus*, and *Terminalia catappa*, in several regions of the world [24,59,63,67]. In some cases, the *C. gattii* complex has also been isolated from other environments, such as soil, air, and water [10,76,77]. Research suggests that periods of higher humidity in temperate climates are more favorable for the proliferation of the *C. gattii* complex; however, different studies on the environmental conditions have reported that the favorable characteristics for the development of this yeast depend on the climatic conditions of the area [60,73,78,79].

Studies published from 2006 to 2016 in different regions of Colombia reported cases of cryptococcosis. The prevalent risk factor reported was HIV infection in 83.5% sufferers and an increase in cases in the female population compared to the previous period of 1997–2005 [78]. Thus, the average annual incidence in the general population is 2.4 per one million inhabitants, while in AIDS patients there is an increase of 1.1 people per 1000 [80]. These data reflect the importance of knowing the distribution of these pathogenic species in the environmental isolates to identify possible sources in which infectious propagules may grow.

The objective of this systematic review is to describe the distribution of environmental isolates of the *C. neoformans* species complex and *C. gattii* species complex in Colombia and their associations with environmental conditions.

## 2. Materials and Methods

A systematic search was carried out in eight online databases: BioMed, DialNet, DOAJ, Medline, PubMed, ScienceDirect, Redalyc, and SciELO, following the Preferred Reporting Items for Systematic Reviews and Meta-Analyses (PRISMA) guidelines, which include identification, selection, and inclusion of the reviewed literature. The years of publication were 1990 to 2020. The descriptors used were: *Cryptococcus* spp., environment, meteorological conditions, and ecology. The articles were selected considering the following inclusion criteria: studies of environmental isolates of *Cryptococcus neoformans* and *Cryptococcus gattii* in Colombia, mainly articles in Spanish and English. Studies of clinical isolates reported in Colombia were excluded. The titles and abstracts of 103 articles were evaluated for relevance; after removing duplicate articles and study reviews, 53 articles underwent full-text review, taking into account environmental variables, microbiological identification methods, and molecular characterization, which were included in the total synthesis of the document.

## 3. Results

Colombia has different thermal floors, producing multiple climatic conditions throughout the national territory. The natural habitat of the *C. neoformans* species complex and *C. gattii* species complex is associated with different environments, from cold and temperate climates to tropical and subtropical climates. In this way, the studies carried out in Colombia aimed to identify the distribution of the infectious agent of cryptococcosis, and thus identify the ecological niche of this fungus.

Of the included studies, five reports showed associations between isolates of *Cryptococcus spp.* and environmental variables. Indeed, Granados et al. and Anacona et al. reported a correlation between high and mean temperature values and recovered isolates of *C. neoformans* [18,81]. Conversely, *C. gattii* serotype C had fewer isolates due to extreme minimum and maximum temperatures according to Granados et al. [81]. Furthermore, a negative correlation with relative humidity was shown for *C. neoformans* serotype A and *C. gattii* serotype C [23,81] in comparison with the favorable influence of humidity in recovered *C. neoformans,* as reported by Anacon 2018 et al. [18]. Additionally, *C. gattii* serotype B exhibited a positive correlation with relative humidity, associated with the high recovery under low evaporation conditions [81].

For the climatic variable precipitation, the *C. neoformans* serotype A evidenced a strong correlation [81]. For solar radiation, Granados et al. and Vallejo et al. reported that *C. neoformans* serotype A and *C. gattii* serotype C exhibited a positive correlation with high solar radiation values, in contrast to *C. gattii* serotype B, which showed a positive correlation with low values of solar radiation [23,81].

Regarding the sample source, Caicedo and Vallejo reported that the accumulation of excreta and the high presence of pigeons result in increases in the recovery of the *C. neoformans* fungus [23,28].

Finally, Caicedo reported that wooden boxes and flat ceilings were the nesting areas where there were more significant isolates of *C. neoformans* [28].

The studies carried out in Colombia confirm the presence of pathogenic species of the genus *Cryptococcus* in the region, with the *C. gattii* species complex being reported in tropical to temperate climates, demonstrating that this species is highly adaptable to different ecological conditions in the country [25,60]. Other isolates of *C. neoformans* have been reported in temperate and tropical climates, associated mainly with avian excreta [18,22,23,56,82]. Environmental isolates of these pathogenic species have been reported in different cities in Colombia, such as Popayán [18], Pasto [23], Cúcuta [26,27], Monteria [19], Bogotá [18,19], and municipalities of Cundinamarca [21,29,61], with Bogotá being the city with the highest number of positive samples [20,81]. These results are discussed throughout the article and are synthesized in Figure 1 and Table 1, where the record of environmental isolates of *C. neoformans* and *C. gattii* in different parts of Colombia are shown.

### 3.1. Atlántico

The city of Barranquilla is the capital of the department of Atlántico, in which a warm climate predominates, with an annual temperature of 28 °C; this region has bimodal rainy seasons that go from May to June and from September to November, with average annual rainfall ranging between 500 and 1500 mm, with dry seasons from December to March and from June to July. Between 2012 and 2014, Noguera et al. conducted a clinical and environmental study, whereby 32 sampling points were recorded, consisting of local parks and areas with a high density of trees. From 1068 environmental samples, 0.4% were positive for *C. neoformans* molecular type VNI in almond trees and roses. The researchers concluded that *C. neoformans*, a molecular type VNI, had a higher prevalence than *C. gattii* and was associated with human exposure in this geographic region [83].

### 3.2. Bogotá

In Bogotá in 1994, Duarte et al. evaluated the association of *Cryptococcus* species with the eucalyptus trees *E. camaldulensis* and *E. terericornis*, distributed in 13 areas located in the northeast, east, west, and center of Bogotá. This city has an altitude of 2630 m above sea level (masl), while at the sampling time the city had a humidity of 65.5% and average temperature of 16.4 °C; the location is shown in Figure 1F. It should be noted that the study did not report the molecular characterization of the isolates. From the results of the 13 evaluated areas, 572 isolates were recovered, of which 27 (4.72%) were of the *Cryptococcus* genus, distributed as follows; 8 were *C. laurentii*; 5 were *C. macerans*; 4 were *C. ater* and 4 were *C. uniguttulatus*; 2 were *C. neoformans*; and 1 each was *C. hungaricus, C. albidus, C. kuetzingii*, and *C. heveanesis*. For the remaining 500 samples, the authors did not report any identification. An important finding reported in this study was the establishment of the leaves as the main sources of isolates for *Cryptococcus* [21].

Another environmental report by Castañeda and Castañeda in 2001 showed the association between isolates of *Cryptococcus* species and *Eucalyptus* trees in a park located in the northeast of Bogotá. No environmental data were recorded during the investigation; however, according to the climatic parameters reported by IDEAM in this period, Bogotá had an average temperature of 13.4 °C and an annual rainfall of 453.91 mm [84,85]. The study did not include a molecular characterization of the isolates. *C. laurentii* was the mainly isolated species, with 17 positive samples, followed by *C. neoformans* var. *neoformans* with two positive isolates, and finally one *C. albidus* isolate. Contrary to what is reported in the literature, where a close association between *C. gattii* and *E. camaldulensis* is described, in this study no isolates of this species were obtained [20].

The study published by Granados and Castañeda showed the results of sampling carried out in 2003 during February to May in 3 areas of Bogotá: The José Celestino Mutis Botanical Garden (JB) (4°41′ N, 74°06′ W), El Lago metropolitan park (PEL) (4°40′ N, 74°03′ W), and the campus of Universidad Nacional de Colombia (UN) (4°38′ N, 74°06′ W). Climatological data for this period showed an average long-term temperature of 19.1 °C, with precipitation being between 500 and 1000 mm [85,86]. In total, 480 samples associated with the trees (form the bark, the soil around the trees, and debris) and 89 samples of bird droppings were collected. The authors did not report molecular characterization for the isolates. Of the 480 samples obtained from the trees, 38 (7.9%) were *C. neoformans*.

It should be noted that in those trees that were inhabited by birds, the frequency of isolation was lower (21.87%) compared to samples isolated from the trees or in decomposing wood (48.75%), which were not frequented by birds. For the statistical analysis, ANOVA, MANOVA, canonical discriminant analysis, Pearson’s correlation, and Chi squared were performed, using the Statistical Analysis System (SAS) for Windows v. 8.02. The frequency of isolation in the excreta samples was 6.7% [87] for *C. neoformans* var. *gattii* and 1% (5) for *C. neoformans* var. *grubii*, which were significantly higher in the dry excrement than the humid excreta. Based on the results, exposure to sunlight does not represent a decisive factor influencing the recovery of yeast. In this study, the fungus showed more frequent colonization in the debris found around trees, supporting the concept that the main niche appears to be the result of the biodegradation of wood. In contrast, the rainy months (April and May), characterized by high rainfall, humidity, few hours of sunlight, and less extreme temperatures (an average temperature of 5 °C higher), favored the appearance and spread of *C. neoformans* in the environment more than the dry months. Finally, this study shows how *C. neoformans* can colonize several zones in a city, and how its population densities can vary within the zones; this distribution could be explained by the capacity of birds to disperse propagules in the wind [81].

In 2010, Escandón et al. collected samples over six months, between February and August 2007. The authors did not report environmental data during the investigation; however, according to the climatic parameters reported by IDEAM in this period, Bogotá had an average temperature of 19.124 °C and precipitation range of 500–1000 mm [86,88]. In total, 28 bark, 37 soil, 52 debris, 4 seed, and 7 flower samples were taken from a total of 91 trees. Isolates were typed using RFLP of the *URA_5_* gene, while PCR was used for the determination of mating types. Of the 128 samples collected, 15 (11.7%) isolates were identified as *C. gattii* molecular type VGIII, mating type a; of these isolates, three were from the extract of the inflorescence of red flowers, while the rest were from the detritus of *Corymbia ficifolia*. Of the 113 remaining samples, the researchers did not specify any identification. The present study reports the isolation of *C. gattii* in *C. ficifolia* trees in Colombia, as represented in Figure 2. This is the first report to reveal this tree species as a habitat for the fungal species [59].

### 3.3. Cali

In 1996, 20 communes of Santiago de Cali were selected as sampling zones for the presence of nesting sites. Santiago de Cali is the capital city of Valle del Cauca department (Figure 1G), which has an average temperature of 28.55 °C and a precipitation range of 1000–1500 mm according to the reports by IDEAM [86,88]. In total, 119 samples of pigeon droppings (*C. livia*) from ceilings, roofs, and wooden boxes were recovered. All samples collected on roofs were negative, while all those obtained from ceilings were positive for *C. neoformans*. Molecular characterization was not performed. The statistical analysis consisted of descriptive statistics and the Chi square test, however the program used was not indicated. In total, 47 % of the samples were identified as *C. neoformans*. Researchers associated high exposure to ultraviolet rays with the absence of *C. neoformans* in roof samples because the sunlight exposure limited the growth of yeast. They also found that where there was a higher number of pigeons, the recovery rate of the fungus was higher (Figure 2). Finally, given the significant increase in cryptococcosis in AIDS patients, the authors noted the potential risk for these patients of acquiring cryptococcosis in urban environments in the city of Cali [28].

In addition, Caicedo et al. in 1996 published a study carried out at the Cali Zoo between 1994 and 1995. The researchers did not report environmental data during the sampling; however, according to the climatic parameters reported by IDEAM in this period, the city of Cali presented an average temperature of 28.55 °C, with precipitation ranging between 1000 and 1500 mm [86,88]. In total, 380 samples were recovered; 110 from bird droppings, 148 from the air inside the cages, and 122 from the air outside the cages. The study reported two isolates of the species *C. neoformans* var. *neoformans*, one (0.9%) from excreta of caracara (*Polyborus plancus*) and the other from a petri dish exposed in the area where the same species of falconiform bird lived (Figure 2); the latter corresponded to 0.3% of the air samples, while the other air sample results were negative. The results obtained in the zoo in the city of Cali showed a low frequency of isolates of this yeast, which was associated with the fact that the cleaning and disinfection procedures that were implemented were adequate [29].

### 3.4. Cúcuta

In 1998, Callejas et al. published a study that aimed to search the *C. gattii* habitat in the city of Cúcuta. The municipal area covers approximately 1176 km^2^. The altitude is 320 masl, the average temperature is 27 °C, and the average annual rainfall is 763 mm. This city is located in the northern region of Santander, as shown in Figure 1C. A total of 157 samples collected from May to September 1997 from 68 trees of *T. catappa*, 90 debris samples, 54 samples from other types of plant materials (38 from seeds, seven from bark, eight from leaves, and one from flowers), and 13 air samples were also collected. The authors did not report on the molecular characterization. The researchers were able to isolate four positive samples for *C. gattii* serotype C (2.54%), corresponding to plant debris; they did not report on the identification for the remaining 153 samples. The researchers concluded that there may be an association between plant dendrites and the positive isolates of *C. gattii*, this being a type of habitat in which yeast is found [25], as shown in Figure 2.

In 2001, Castañeda et al. published a study that sought to determine the association of the presence of *C. gattii* (at that time classified as *C. neoformans* var. *gattii*) with the species *T. catappa* (almond trees). The authors reported on a total of 370 samples, of which 160 were from almond debris and 210 were from 9 new almond trees, selected from January 1998 to July 1999. The study reported that 31 samples (19.3%) were positive for *C. gattii* serotype C (formerly *C. neoformans* var. *gattii*) of the 160 samples initially collected in the study, while of the 210 samples collected from the nine additional almond trees, only 1 (0.48%) was positive for *C. gattii*. The authors established that it was not possible to isolate the *C. neoformans* species complex from any of the almond trees, thus suggesting that there is a specific association of *C. gattii* with this tree species [24]. Another environmental report was made in 2011 in Cúcuta by Firacative et al. This area has an average temperature of 28 °C and an approximate average annual rainfall of 763 mm. In total, 4389 samples were processed, among which samples of soil, bark, debris, leaves, and fruits were collected, which were associated with 3634 trees from different species, including *T. catappa* (almond), *Ficus* sp. (Higuerón), *Licania tomentosa* (Oiti), *Pithecellobium dulce* (Chiminango), *Fagara rhoifolia* (Tachuelo), *Melicoccus bijugatus* (mamón), *Enterolobium cyclocarpum* (piñon), and *Mangifera indica* (mango) (Figure 2). Molecular typing of isolates was performed by PCR fingerprinting and RFLP of the *URA_5_* gene. From 4389 samples, only six were positive—three (0.07%) for *C. gattii* and three (0.07%) for *C. neoformans*. From *C. gattii-*positive samples, seven isolates were C serotype, VGIII molecular type; and one was serotype B, VGI molecular type. Of the positive samples for *C. neoformans*, 13 isolates were classified as a *C. neoformans* var. *grubii*, serotype A, VNI molecular type. All recovered isolates were from the soil samples. Contrary to expectations for the study area, which has a high rate of cryptococcosis due to *C. gattii*, there was a low rate of environmental recovery for this species, suggesting the importance of establishing environmental variables that directly affect the survival of microorganisms in the environment, meaning samplings should be taken at specific times of the year [26].

In 2019, Angarita et al. carried out a study in the city of Cúcuta to identify and characterize *Cryptococcus* environmental isolates in ten public areas. The study did not report environmental data for the months of sample collection; however, as reported by IDEAM for this period, the average temperature was 24.95 °C, with annual rainfall ranging between 50 and 74 mm [85]. A total of 1300 samples from 446 trees belonging to 10 different species were found, including soil (442), bark (434), nuts (40), and leaves (384). PCR fingerprinting and RFLP of the *URA_5_* gene were used to determine the molecular pattern. The results obtained in this study established the presence of *C. neoformans* in 19 trees (4.3%) and *C. gattii* in one tree (0.2%), for a total of 21 isolates and 20 positive trees. Santander Park, with 47.6% (10 out of 21) of the isolates, was the area that presented the most significant number of positive samples, followed by La Victoria Park with 23.8% (5 of 21 isolates), all of which were identified as *C. neoformans.* Using a different method, *C. gattii* was isolated from the *Ficus benjamina* tree located in the Mercedes Ábrego park (Figure 2). Genotypic analysis revealed the presence of *C. neoformans* var. *grubii* VNI molecular type in 85.7% of the environmental isolates, followed by VNII in 9.5% and *C. gattii* VGII molecular type in 4.8% of the isolates. This study revealed the importance of performing longitudinal sampling of the environmental niches reported in the literature for this fungus, thus allowing its presence to be established in the sampled areas, suggesting this type of sampling as a tool for permanent surveillance of this pathogenic fungus in endemic areas of the city of Cúcuta [27].

### 3.5. Cundinamarca

Cundinamarca department is in the central region of the country, located between 3°42′ and 5°51′ latitude north and 73°03′ and 74°54′ longitude west of the Greenwich meridian. The altitude is between 2000 and 3000 masl and the temperature ranges between 12 and 18 °C, with a predominant cold thermal floor. The Páramo regions have temperatures equal or less than 12 °C and are located at an elevation above 3000 masl. Quintero et al. carried out a study in different areas of Cundinamarca in order to evaluate under which climatic conditions the *Cryptococcus* species are best adapted, taking into account the topographic, natural, and climatological diversity of the department. Of the 765 samples processed, 437 corresponded to eucalyptus debris, 182 to almond debris, and 146 to pigeon droppings. The results indicated that *C. neoformans* was present in 12 (46%) of the municipalities studied, indicating that 104 (14%) samples were positive, whereby the pigeon droppings (21%) were the principal substrate. A total of 92 isolates (88%) were recovered from the cold thermal floor. In the Páramo area, the yeast was not isolated due to the environmental conditions, such as the precipitation and high relative humidity, few hours of sunlight, and temperate thermal floor between 18 and 24 °C, which favor the propagation of this yeast in the environment, as well as the high altitudes of between 1000 and 2000 masl. In conclusion, the authors confirmed the association between *C. neoformans* var. *grubii* serotype A and pigeon feces, in addition to confirming eucalyptus as an environmental niche for *C. gattii* serotypes B and C, while almonds were also established as possible hosts for serotype C (Figure 2). Finally, the importance of the cold thermal floor as a habitat for the fungus was established [56].

However, a study carried out by Escandón P. in the municipality of La Calera (Cundinamarca) from February to March of 2003 evaluated the presence and distribution of species of the *C. neoformans* complex, especially serotype B, which were associated with the study area. This region has two thermal floors, namely cold and temperate, in addition to having an altitude of 2728 masl, with an average temperature of 12.7 °C and annual precipitation of 250 mm. The researchers recovered 195 samples from trees debris, 167 from *Eucalyptus* spp., and 28 from other species, such as pines (*Pinus s*p.), white snakeroot tree (*Ageratina altissima*), Laurel (*Laurus nobilis*), Raque (*Vallea stipularis*), and Guaba (*Phytolacca bogotensis*). Of the 167 samples taken with a swab, *C. gattii* was found in February in 3 out of 40 samples (7.5%), on March 6 32 out of 40 samples (80%), and on March 12 in 11 out of 87 samples (12, 5%). Additionally, *C. gattii* serotype B was recovered from 46 of the 167 samples of *Eucalyptus* (27.5%). The presence of yeasts with small capsules was confirmed in 30 (65.2%) of these, while the remaining 16 were strains without capsules. This study was the first report on *C. gattii* serotype B. It should be noted that the fungus was found on different trees species, as observed in Figure 2 [60,89].

### 3.6. Huila

In 2018, Virviescas B. et al. performed a study in the city of Neiva to determine the distribution of the etiological agents of cryptococcosis in the region. The municipality has an average annual temperature of 27.7 °C, is located at a latitude of 2°59′55” north and a longitude of 75°18′16” west, and has an altitude of 442 masl, with an approximate population of 345,911 inhabitants. In total, 118 samples were collected—32 from the fruits, leaves, and debris of almond trees (*T. catappa*), and 86 from pigeon droppings (*C. livia*). The samples were processed using conventional techniques. Species were determined by seeding on Canavanine-Glycine-Bromothymol Blue (CGB) agar, and molecular typing was performed using PCR fingerprinting and RFLP of the *URA_5_* gene. Of the total samples processed, 40 colonies obtained from eight (6.8%) samples were identified as *C. neoformans* var. *grubii* and isolated from bird droppings, which can be observed in Figure 2, contrary to tree samples, where no positive samples were recovered. Likewise, *C. gattii* was not isolated from the collected samples [22].

### 3.7. Monteria

Between 2008 and 2009, an investigation was carried out by Contreras et al. in Monteria, the capital of the department of Córdoba (Figure 1D), located in northern region of Colombia, to determine the ecological association of *C. gattii* with almond trees (*T. catappa)* present in the urban area of the city. Monteria has an altitude of 20 masl, a warm climate with temperatures ranging between 28 and 35 °C, a relative humidity of 85%, and pluvial precipitation of 1200 to 2500 mm annually. The researchers collected 2445 samples from 163 *T. catappa* trees. The samples were collected, taking into account the parts of the bark with grooves and hollows. The study did not include molecular characterization. The statistical analyses included non-parametric analysis, Tukey’s hypothesis test, and Chi-squared tests using Statgraphics Centurion version 15.2.

Of the total samples processed, *C. gattii* was isolated from 217 samples, which corresponded to 8.9%. The authors highlighted that the sample collection corresponded to a rainy period involving more humidity and a higher accumulation of debris. Given the obtained results, the researchers consider almond trees an ecological niche for *C. gattii* (mainly the fruits, which are essential), since in the region the ripe fruit is edible and could indicate a point of spread for the fungus. These findings were the first report of the isolation of *C. gattii* from environmental sources in the department of Córdoba [19].

### 3.8. Pasto

For the municipality of Pasto, Vallejo et al. in 2016 reported a study that carried out an analysis of the residential and populated areas with a high density of pigeons. The study did not report environmental data; however, according to the climatic parameters reported by the IDEAM, Pasto has an altitude of 2527 masl, an average temperature of 19.2 °C, and a maximum average annual rainfall of 106,267 mm [85]. The city is located on the Andes cordillera, as shown in Figure 1B. In total, 128 samples were recovered from the different areas; most of the samples were from feces without debris, some contained residues of soil and vegetables, while others contained feathers. The samples were exposed continuously to light and variably to the environment. Sampling areas were characterized and classified as high, medium, and low, according to the environmental conditions of each area and the characteristics of the samples.

Contrary to the procedure reported by Caicedo et al. [28], samples were exposed to long hours of sunlight, which is an excellent condition for obtaining positive samples, as it is different to the low humidity in the samples, which affects the recovery of the fungus. The study also showed that a higher number of pigeons per nest favored a higher percentage of isolates of the fungus being obtained [23]. The study did not perform molecular characterization of the obtained isolates. The fungus was isolated in 9 out of 10 sampled areas, whereby 26.5% of the total samples collected corresponded to positive isolates of *C. neoformans*. The number of isolates was lower in fresh samples, while wet samples (not necessarily fresh) provided a higher number of positive isolates, similar to the results reported by Granados and Castañeda in 2006 [90]. Descriptive statistics and the Chi-squared test were used for statistical analysis using SPSS Statistics 20.0 software under shareware license. This study established that of the pigeon droppings samples taken in urban areas, the prevalence of C. neoformans was 90%, demonstrating an association of the avian species with *Cryptococcus* isolates (Figure 2). Additionally, the importance of establishing the variables was confirmed, such as the exposure to sun or shade, and of checking whether the avian excreta were fresh, wet, or contaminated with plant residues, which could be associated with an increased probability to obtain isolations positive for this yeast [23].

### 3.9. Popayán

Popayán is the capital of the Colombian department of Cauca, with an elevation of 1735 masl, a subtropical highland climate with an annual average temperature of 18–19 °C, with maximum temperatures occurring from July to September (high temperature of 29 °C and minimum temperature of 10 °C) and a precipitation of 1941 mm. The city is located on the west of Colombia, as shown in Figure 1A in the distribution map. A study reporting environmental isolates was published in 2018 by Anacona et al.; they collected a total of 303 samples of bird droppings from pigeons (*C. livia*) and herons (*Bubulcus ibis*) between September 2012 and June 2013. These associations can be observed in Figure 2.

The authors collected and processed the samples using conventional techniques. The identification of presumptive isolates of the *C. neoformans* species complex was also carried out by means of conventional procedures, along with the molecular characterization of the isolates. One-way ANOVA, post hoc analysis, Tukey’s test, and the Chi-squared test were performed using SPSS version 15 for statistical analysis. A total of 118 (38.94%) samples were positive for *Cryptococcus neoformans* var. *grubii*, with 99.2% corresponding to the VNI molecular pattern and the remaining 0.8% to VNII. Contrary to the report for the pigeon excreta samples, no species of the *C. neoformans* species complex was isolated from B. ibis. In this way, the study allows us to confirm that bird droppings, especially of *C. livia*, are a fundamental environmental niche for *C. neoformans* var. *grubii* molecular type VNI, since they allow growth and reproduction of the fungus, representing a potential source of infection for the susceptible population in Popayán [18], as shown in Figure 2.

## 4. Discussion

The *Cryptococcus* genus comprises a group of fungal pathogens that frequently affects humans; without timely and adequate treatment, it can be fatal. In order to establish control measures for infectious diseases, it is highly relevant to provide a report on the fungal environmental distribution throughout a country.

According to the environmental studies reported in Colombia until 2019, climatic conditions are essential when analyzing the requirements that favor the appearance of the fungus (Appendix A). The studies report an extensive distribution of the fungus at different altitudes ranging from 20 masl (e.g., Monteria city [19]) up to 3000 masl (e.g., Cundinamarca department) [54]. The fungus can survive in temperatures as low as 12 °C, as demonstrated by Quintero [56], or in high temperatures reaching 28.8 °C, as reported by Cúcuta [24], thus demonstrating its wide distribution range in the country, as shown in Figure 3.

Regarding the numbers of samples collected for the isolation of *Cryptococcus* spp., it can be said that the most significant number of samples collected was in the city of Cúcuta at 4338 samples, of which only 0.14% were positive, which may be due to the temperatures at this site (27.8 °C to 28.8 °C) [24]. This low frequency of positive isolates is in contrast with the results found in the city of Popayán, where the numbers of samples collected was 303, 38.94% of which were positive. This city has a temperature range of between 10 and 29 °C [18]. In the same way, this region was established by Mak et al. as an optimal ecological niche for isolation of *C. neoformans* species [89]. Similarly, studies carried out in the department of Cundinamarca resulted in 13.59% of positive samples for *Cryptococcus* from the 765 samples collected under temperature conditions ranging between 12 and 18 °C [56]. The frequency of positive *Cryptococcus* isolates is higher in environments with low temperatures, since such conditions promote their conservation [91].

The methods used for the identification and characterization of the *Cryptococcus* species complex have mainly involved extraction with saline phosphate buffer and identification using *G. abyssinica* seed agar. In some studies, Indian ink staining was used to identify the fungus capsule, being a simple and inexpensive method. The molecular methods generally used in the studies were restriction fragment length polymorphism (RFLP) of the *URA_5_* gene and PCR fingerprinting with primer (GTG)_5_.

Data analysis was not performed in most of the studies, except when reporting the percentages of positive samples found for each species according to the nomenclature used by each study. However, some studies have reported statistical data analysis, including Granados et al. in Bogotá [81], Caicedo et al. in Cali [28], Quintero et al. in Cundinamarca [56], Contreras et al. [19], Vallejo et al. in Pasto [23], and Anacona et al. in Popayán [18]. According to the results of these statistical analyses, the climatic variables that influence the recovery of the fungus are precipitation, temperature, evaporation, solar radiation, and humidity (Appendix A).

The studies carried out in Colombia, as with the reports worldwide, demonstrate the association of the *C. neoformans* complex with excreta of *C. livia* pigeons and of some *C. neoformans* species with *P. plancus,* thus affirming the association of the fungus with different avian species. Bird droppings contain xanthine, urea, uric acid, and creatinine, substrates that the fungus can assimilate within [11,31], given the presence of enzymes such as creatinine that favor the adaptation of yeast to this type of environment [92]. This proves that bird droppings are a secondary and temporary niche for this species [55].

In addition, the probability of finding *Cryptococcus* is higher if the samples are wet, as shown by Quintero et al. In Cundinamarca region, 88% of the isolates were recovered from the cold thermal floor, with high precipitation and relatively humid environmental conditions, which favored the occurrence and spread of the fungus in the environment, as reported by Cogliati et al., who found a higher occurrence of the fungus in rainy seasons or wet periods [53,93]. Similar to the Cundinamarca study, a study carried out in Australia established that yeast could be isolated from eucalyptus debris and pigeon droppings, however the study had a particular focus on cold seasons with high relative humidity [54,94]. Contreras and collaborators in 2011 reported similar data, where they highlighted that the highest percentage of positive samples was because the collection was performed in rainy seasons, which promoted more significant accumulation of debris [19]. This was in line with the study by Vallejo et al., who established that wet samples contain higher numbers of isolates of *Cryptococcus* species.

Vallejo et al. in 2016 obtained positive samples for the *Cryptococcus* species complex in places exposed to constant sunlight, thus relating the amount of brightness to the presence of the fungus [22], since the ability to produce melanin not only helps the fungus resist solar radiation, but it can also use it as metabolic energy [23,95]. This aspect differed significantly from the rest of the studies, which mentioned that the solar brightness received by the fungus is a factor that prevents its growth. For example, in the study carried out in Cali, the authors did not collect positive samples from places with high sun exposure [29].

According to the samples registered for the different studies, the presence of the *C. neoformans* complex is generally associated with debris from trees of the genus *Eucalyptus* and *Ficus* sp., as stated by Lazéra et al. [96] and Cogliati et al., who found associations with arboreal specimens as niches and stable reservoirs [53,93]. Similar results were also found in the regions of Popayán, Cundinamarca, Bogotá, and Cali (Figure 4). Likewise, this species is strongly associated with avian droppings, generally from pigeons (*C. livia*) [87], which is consistent with the findings described Kwon-Chung and Bennett in 1984 [47], who established the primary environmental source of *C. neoformans* as being pigeon droppings, which was contrary to the findings reported by Cogliati et al., who that suggested that poultry feces is a secondary and transitory niche [53,93].

*C. gattii* has been found in different regions, such as Cúcuta and Monteria, which have average temperatures of around 27 °C, as well as Cundinamarca, which has an average temperature of 13 °C. In agreement with the study by Cogliati et al., where identified that warm temperatures favors the distribution of *C. gattii* complex [53,93]. The trees species associated with this yeast are *T. catappa* and *Ficus sp.*, as reported by Castañeda et al. [24] and corroborated by the study by Escandon et al., in which the survival of the *C. gattii* complex was shown for up to 12 months in these trees species [97]. Additionally, Firacative et al. recorded positive isolates from the soil close to the tree species *Ficus sp.* [26]. In the city of Monteria, Contreras et al. reported *C. gattii* in soil samples from *Ficus spp.* [19]. This association between the *C. gattii* complex and rubber and almond species has been reported in different parts of the world. For example, the study by Randhawa et al. in 2003 reported decomposed wood in *Ficus religiosa* trees as an environmental source of *C. gattii* [73]. These findings indicate that these tree species could be a natural habitat for *C. gattii* complex [94], describing them as reservoirs for this yeast species [65,97,98]. The *C. gattii* complex is difficult to isolate from the environment [99,100]. However, the findings from a few studies showed high numbers of positive samples for this species. For the *C. neoformans* complex, the number of positive isolates reported in each study was low, however the number of studies is much higher [19,26,27], as can be seen in Figure 4.

Both the *C. neoformans* complex and the *C. gattii* complex have been reported in clinical isolates; however, the *C. neoformans* complex is mainly isolated from the environment, indicating the presence of the two species in the environment [101]. Thus, by expanding the search for the *C. gattii* complex in the environment, the number of studies has increased, with the largest numbers of isolated samples being obtained for the species of *C. gattii*, especially serotype C in Cúcuta, which has a high incidence of *C. gattii* serotype B in immunocompetent patients [19,24,25,26,27]. In contrast, in La Calera, Cundinamarca, species of *C. gattii* with serotype B have been isolated in low proportions [60]. The two serotypes are considered primary pathogens, which therefore infect immunocompetent individuals, showing the need to establish the foci of infection for this species [102].

Regarding genotypes, *C. gattii* VGII is the most common in clinical isolates, while the VGIII genotype is less frequent [102]. However, in the study by Escandon et al., the most common phenotype determined in environmental isolates was VGIII, in agreement with the results reported by Firacative et al. in 2011, although to a lesser extent [26,57]. Additionally, the predominant genotype for clinical isolates of *C. neoformans* is VNI [102,103], which coincides with the genotype determined in environmental samples [18,26,27].

Finally, it should be mentioned that as in other regions, in Colombia the soil samples represented a poor environment for the recovery of the *C. neoformans* species, while the excreta of birds was considered an important reservoir for this species [73,99,100].

## 5. Conclusions

The studies carried out in Colombia have shown that the *C. neoformans* species complex and *C. gattii* species complex share some niches in terms of where they are established, being distributed in regions ranging from 20 to 3000 masl. Similarly, positive isolates were found in cold and temperate climates, with low sunlight and high relative humidity, thus showing that in wet samples, the probability of finding species of the genus *Cryptococcus* is higher. In contrast, although to a lesser extent, it has been established that in some cases these yeast species can grow in areas with a significant influence of sunlight, which may be related to the production of melanin as a protective factor against UV rays, giving this species adaptability to these environments [89] In this way, it is essential to investigate the production of this virulence factor in the isolates that recover under conditions of a strong influence of solar brightness in future studies.

Furthermore, the methods used for isolation and identification of the *Cryptococcus* complex species, such as *G. abyssinica* agar, urease test, Indian ink staining, CGB agar, RFLP of the *URA_5_* gene, and PCR fingerprinting with primer (GTG)_5_, correspond to the methods mostly used in Colombian studies. It is worth highlighting their effectiveness for the characterization of these two species.

Finally, the reviewed studies allowed us to establish that the most significant numbers of *Cryptococcus* spp. species were isolated in urban areas, such as squares, parks, churches, and zoos, which could be related to human activity, therefore increasing the possibility of infection through these sources.

## Figures and Tables

**Figure 1 jof-07-00410-f001:**
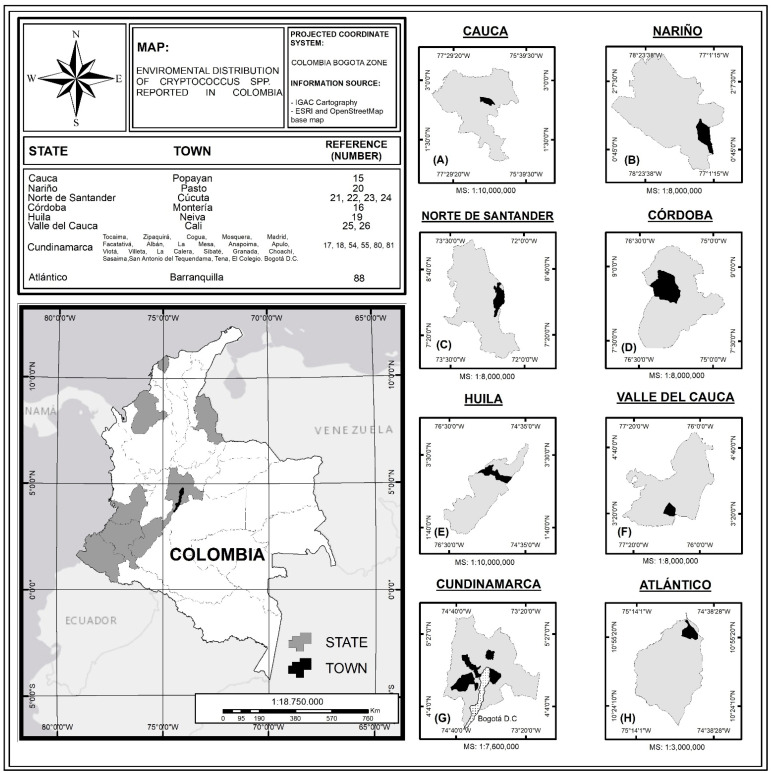
Distribution of environmental isolations of *Cryptococcus* spp. in Colombia; MS: Map Scale. (**A**) Department of Cauca, (**B**) Department of Nariño, (**C**) Department of Norte de Santander, (**D**) Department of Córdoba, (**E**) Department of Huila, (**F**) Department of Valle del Cauca, (**G**) Department of Cundinamarca, and (**H**) Department of Atlántico. Source: The authors of this study, based on the information recorded in Table 1.

**Figure 2 jof-07-00410-f002:**
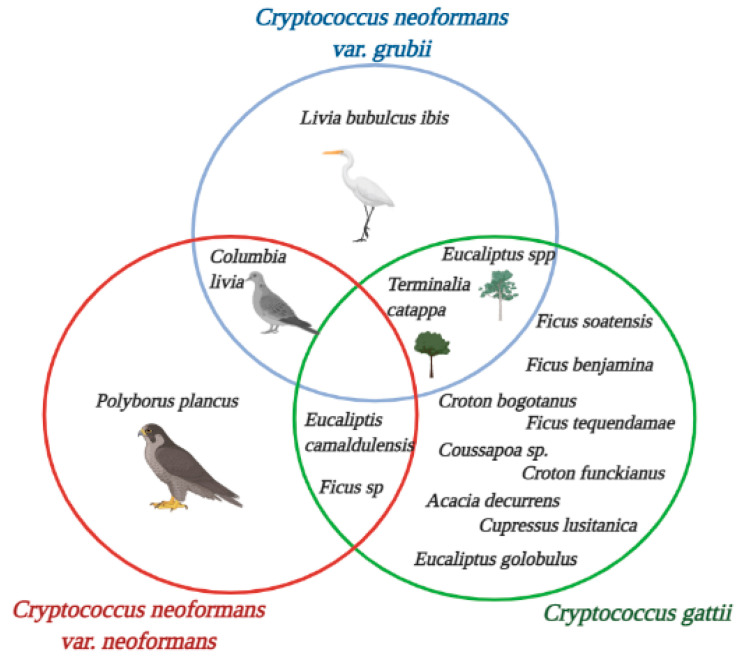
Environmental isolate sources of *C. neoformans* and *C. gattii* species complexes in Colombia. Source: The authors of this study; image created in BioRender.com.

**Figure 3 jof-07-00410-f003:**
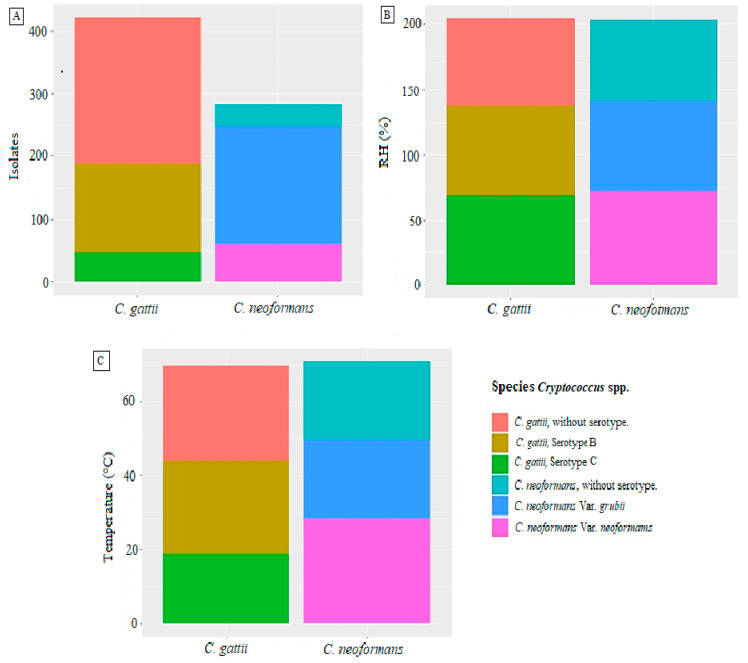
Environmental conditions reported for each species of *Cryptococcus* spp. (**A**) Numbers of isolates reported for each species *(**C. gattii* and *C. neoformans*), according to their respective varieties. (**B**) Percentages of relative humidity (RH) reported for the isolates of each species (*C. gattii* and *C. neoformans*), according to their respective varieties. (**C**) Temperatures (°C) reported for the isolates of each species (*C. gattii* and *C. neoformans*), according to their respective varieties. Source: The authors of this study; image created in RStudio.

**Figure 4 jof-07-00410-f004:**
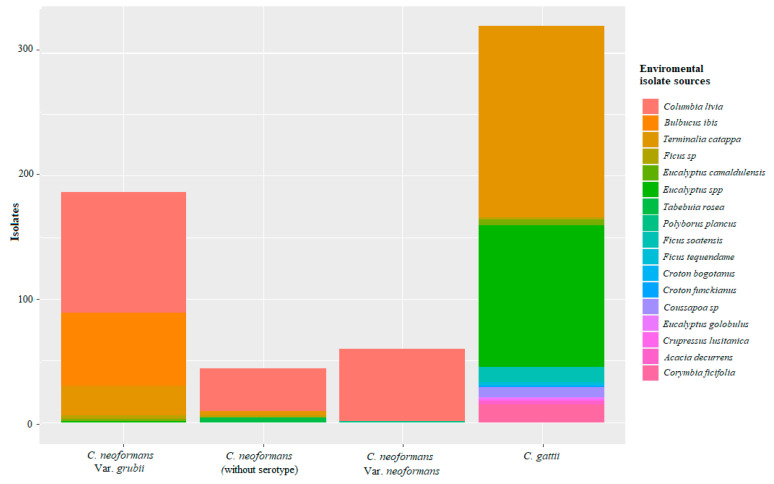
Isolations reported according to their environmental source for the complexes of species of *C. neoformans* and *C. gattii* in Colombia. Source: The authors of this study; image created in RStudio.

**Table 1 jof-07-00410-t001:** Environmental isolates of the *Cryptococcus* genus registered in Colombia.

Reference (Number)	City	Sampling Period	Total of Samples		Climatic Conditions	Isolated Species		Type of Samples	
			Number	Positive Environmental Isolated (%)		Species	n	Samples	Samples Species
[83]	Barranquilla	September 2012–November 2014	2068	0.4	T (°C): 28 P (mm): 500–1500	*C. neoformans*	8	Leaves, flower, land, cortex, dendrites	*Almendro (Terminalia catappa),* *Tabebuia rosea*
[20]	Bogotá	1997–1999	426	3.99	T (°C): 13.4 *	*C. laurentii*	14	Bark	*Eucalyptus camaldulensis*
						*C. neoformans* var*. neoformans,* serotype A	2		
						*C. albidus*	1		
[21]	Bogotá	1994 (August and November)	562	4.72	T (°C): 16 RH (%): 62.5	*C. laurentii*	8	Flower (2), seeds (2), leaves (1), stems (1), debris (2)	*Eucalyptus sp.*
						*C. macerans*	5	Flower (2), leaves (2), stems (1)	
						*C. gattii serotype C*	4	Seeds (1), leaves (2), stems (1)	

[81]	Bogotá	2003 (January–May)	480	7.92	T(°C):19.124 500 and 1000 mm *	*C. neoformans var. gattii*	33	Bark (23), soil (12), and hollow (3)	*Ficus soatensis (11), Ficus tequendamae (2), Croton bogotanus (1), Croton funckianus (1), Coussapoa sp. (1), Eucalyptus camaldulensis (7), Eucalyptus golobulus (3),* *(2), Cupressus lusitanica (1), Acacia decurriiiiens (3)*
						*C. neoformans var. grubii*	5		
[59]	Bogotá	2007 (February and August)	128	20	T(°C): 19.124 500 and 1000 mm *	*C. gattii*	15	Bark (28), soil (37), debris (52), seeds (4), and flowers (7).	*Corymbia ficifolia*
[28]	Santiago de Cali	-	119	49.58	T (°C): 28.55 *	*C. neoformans* var. *neoformans*	59	Pigeon feces	*Columba livia*
[29]	Santiago de Cali	1994–1995	380	1.6	pH 6.0	*C. neoformans* var. *neoformans*	1	Bird droppings	*Polyborus plancus*
[25]	Santiago de Cúcuta	1997 May–September	157	12.546	P (mm): 763	*C. gattii serotype C*	4	Debris	*Polyborus plancus* *Terminalia catappa*
[24]	Cúcuta	1997–1999	370	19.38	T (°C): 28.83 RH (%): 69 P (mm): 595–1341	*C. gattii,* serotype C	31	Debris	*Terminalia catappa* (almond-tree)
[26]	Cúcuta	2008 (January, February and August)	4389	0.14	T (°C): 28.8 (RH) (%): 68	*C. gattii* (serotype C)	2	Soil around the trees	*Ficus sp.*
						*C. gattii* (serotype B)	1		
						*C. neoformans* (Serotype A)	3	Soil around the trees	*Ficus sp.*
[27]	Cúcuta	October 2016–April 2017	1300	4.3	T (°C): 24.95 P (mm): 70–64 * -	*C. neoformans var, grubii*	19	Soil, Bark, nuts and leaves.	*Mango (Mangifera indica),* *Oití (Licania tomentosa),* *Samán (Samanea saman), Tamarindo (Tamarindus indica), Totumo (Crescentia cujete), Chiminango (Pithecellobium dulce) Limón swingle (Swinglea glutinosa), Mamoncillo (Melicoccus bijugatus) y Almendro (Terminalia catappa)*
[56]	Cundinamarca	-	765	13.59	T (°C): 12–18	*C. neoformans,* serotype A	32	Pigeon feces (31), debris (1)	*Columba livia, Eucalyptus spp.*
						*C. gatti* serotype B	62	Debris	*Eucalyptus spp.*
[60]	La Calera	2003 (February and March)	167	27.54	T (°C): 13	*C. gattii* serotype B	46	Debris	*Eucalyptus spp.*
[22]	Neiva	2018	118	6.8	T(°C):27.7 442 masl	*C. neoformans var. grubii*	8	Pigeon feces (8) Fruits, leaves, and debris of almond trees (0)	*Columba livia* *Terminalia catappa*
[19]	Montería	2008–2009	2445	8.88	T (°C): 27	*C. gattii*	217	Flowers, bark and soil	*Terminalia catappa*
[23]	Pasto	-	128	26.56	Considerable humidity and sunshine conditions	*C. neoformans*	34	Pigeon feces	*Columba livia*
[18]	Popayán	2012–2013 (September–June)	303	38.94	T(°C):18–191735 masl	*C. neoformans* var. *grubii*	118	Bird droppings	*Columba livia, Bubulcus ibis*

T: Temperature; P: Precipitation; RH: Relative humidity. *: Data from Institute of Hydrology, Meteorology, and Environmental Studies (IDEAM); -: No information; masl: meters above sea level.

## Data Availability

Not applicable.

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
