# Peer review of "Environmental Status of Cryptococcus neoformans and Cryptococcus gattii in Colombia"

_jof, 2021, doi:10.3390/jof7060410_

Round 1

Reviewer 1 Report

The paper attempts to review the environmental occurrence of pathogenic Cryptococcus species in Colombia, with the aim of assessing the risk for the disease through exposure to some natural environments. Unfortunately, the paper has important flaws that lead me to recommend its rejection for publication in the Journal of Fungi.

The most relevant aspects for this decision are the following:

- Lack of originality: There are at least two excellent papers that previously evaluated the global environmental occurrence of cryptococcus in Colombia. These are references 86 and 87. The work of Granados and Castañeda, published in 2006, which evaluated the influence of climatic conditions on the isolation of yeasts from environmental sources [Granados DP, Castañeda E. Influence of climatic conditions on the isolation of members of the Cryptococcus neoformans species complex from trees in Colombia from 1992–2004. FEMS Yeast Res. 2006;6(4): 636–644]. And the excellent ecological approach made from data collected in Colombia, which appeared in: Mak S, Vélez N, Castañeda E, Escandón P, Group CES. The fungus among Us: Cryptococcus neoformans and Cryptococcus gattii ecological modeling for Colombia. J Fungi 2015; 332-344 I really think that the work presented does not provide any new or interesting information to add to that of these other two.

- The format of the review: The results are presented as a series of summaries of the reviewed papers for each geographical area of the country, without an adequate analysis of the findings based on well-defined variables and using appropriate statistical tools. The methods should define the variables that will be taken in account for the analysis of data and, the analysis should be adequately performed by statistical tools for quantitative and qualitative parameters. On the contrary, you cannot reach solid conclusions.

- The use of incorrect and outdated information, especially on the taxonomy of the genus and the current knowledge of the distribution of these yeasts worldwide and in different environments. The taxonomy of the genus has been revised and deeply modified in recent years based on new knowledge about its genotypes, phenotypes, ecological and clinical aspects. In fact, since 2017 there are two branches of researchers discussing the taxonomy within the two pathogenic Cryptococcus species complexes (as they are no longer single species). The two trends accept the existence of the C. neoformans and C. gattii species complexes with several different strains or species within the complex. As the differences between the yeasts constituting the two groups have much to do with their environmental distribution and pathogenicity, it is of notable importance to take this into account for the analysis intended in this manuscript. I recommend reading the following publications: 

Hagen F, Khayhan K, Theelen B, Kolecka A, Polacheck I, Sionov E, Falk R, Parnmen S, Lumbsch HT, Boekhout T. Recognition of seven species in the Cryptococcus gattii/Cryptococcus neoformans species complex. Fungal Genet Biol. 2015 May; 78:16-48. Epub 2015 Feb 23.

Hagen F, et al. Importance of Resolving Fungal Nomenclature: the Case of Multiple Pathogenic Species in the Cryptococcus Genus. mSphere. 2017 Jul-Aug; 2(4): e00238-17. doi: 10.1128/mSphere.00238-17

Kwon-Chung KJ, Bennett JE, Wickes BL, Meyer W, Cuomo CA, Wollenburg KR, Bicanic TA, Castañeda E, Chang YC, Chen J, et al. The Case for Adopting the "Species Complex" Nomenclature for the Etiologic Agents of Cryptococcosis. mSphere. 2017 Jan-Feb; 2(1). Epub 2017.

Regarding the current knowledge of these yeasts in terms of genotypes, phenotypes, ecological and clinical aspects, since the major outbreak of C. deuterogattii cryptococcosis that occurred on Vancouver Island (British Columbia, Canada) more than 20 years ago (in 1999), many papers have been published demonstrating the presence of C. gattii complex species in temperate areas. Earlier ideas of tropical and subtropical distribution are now not as well supported. In addition, Eucalyptus is one of hundreds of tree species that have been shown to be ecological niches for different Cryptococcus species. In this regard, some reviews to update the information on pathogenic cryptococci and their ecology are very interesting. I recommend the following:

Robin C. May, Neil R.H. Stone, Darin L. Wiesner, Tihana Bicanic, Kirsten Nielsen. Cryptococcus: from environmental saprophyte to global pathogen. Nat Rev Microbiol. 2016; 14(2): 106–117. doi: 10.1038/nrmicro.2015.6

Carolina Firacative, Jairo Lizarazo, María Teresa Illnait-Zaragozí, Elizabeth Castañeda, Latin American Cryptococcal Study Group. The status of cryptococcosis in Latin America. Mem Inst Oswaldo Cruz. 2018; 113(7): e170554. doi: 10.1590/0074-02760170554

Massimo Cogliati, Roberta D'Amicis, Alberto Zani, Maria Teresa Montagna, Giuseppina Caggiano, Osvalda De Giglio, Environmental distribution of Cryptococcus neoformans and C. gattii around the Mediterranean basin. FEMS Yeast Res. 2016 Jun 1; 16(4): fow045. doi: 10.1093/femsyr/fow045

Jorge G. Carvajal, Alberto J. Alaniz, Mario A. Carvajal, Emily S. Acheson, Rodrigo Cruz, Pablo M. Vergara, Massimo Cogliati. Expansion of the Emerging Fungal Pathogen Cryptococcus bacillisporus Into America: Linking Phylogenetic Origin, Geographical Spread and Population Under Exposure Risk. Front Microbiol. 2020; doi:

Massimo Cogliati, Erika Puccianti, Maria Teresa Montagna, Antonella De Donno, Serdar Susever, Cagri Ergin, et al. Fundamental niche prediction of the yeast pathogens Cryptococcus neoformans and Cryptococcus gattii In Europe.  Environmental Microbiology 2017; 19: 4318-4325. DOI: 10.1111/1462-2920.1391510.3389/fmicb.2020.02117

Author Response

Response to Reviewer 1 Comments

The paper attempts to review the environmental occurrence of pathogenic Cryptococcus species in Colombia, with the aim of assessing the risk for the disease through exposure to some natural environments. Unfortunately, the paper has important flaws that lead me to recommend its rejection for publication in the Journal of Fungi.

Point 1: Lack of originality: There are at least two excellent papers that previously evaluated the global environmental occurrence of cryptococcus in Colombia. These are references 86 and 87. The work of Granados and Castañeda, published in 2006, which evaluated the influence of climatic conditions on the isolation of yeasts from environmental sources [Granados DP, Castañeda E. Influence of climatic conditions on the isolation of members of the Cryptococcus neoformans species complex from trees in Colombia from 1992–2004. FEMS Yeast Res. 2006;6(4): 636–644]. And the excellent ecological approach made from data collected in Colombia, which appeared in: Mak S, Vélez N, Castañeda E, Escandón P, Group CES. The fungus among Us: Cryptococcus neoformans and Cryptococcus gattii ecological modeling for Colombia. J Fungi 2015; 332-344 I really think that the work presented does not provide any new or interesting information to add to that of these other two.

Answer 1: Thank you very much for your work on our manuscript. Granados and Castañeda, published in 2006, is an excellent article that includes isolates from 1992 to 2004 in four cities in Colombia. We included it in our review, given the importance it represents in establishing the distribution of members of the Cryptococcus neoformans species complex. However, what we intend with our review article is to describe the works carried out in Colombia, where the different sources of isolation of this yeast between 1999 to 2020 are evidenced and included other cities (n = 10) that were not previously included in the studies of Granados DP and Mark S.

For example, Anacona C et al. reported the first isolation of Cryptococcus neoformans var. grubii in bird droppings in the municipality of Popayán. This region is not reported in any of the mentioned articles but corresponded an optimal ecological niche for C. neoformans according to the niche model established by Mak S et al. 2015. Similar to report by Virviescas B et al. in 2018 and Vallejo et al. 2016, where C. neoformans was isolated in a region characterized as optimal ecological niches. Likewise. Angarita-Sánchez et al. 2019, reported isolates from 9 tree species in an area, also characterized as an optimal ecological niche; of these tree species, 7 constitute the first report in Colombia, a relevant aspect that meets in this review article.

In this way, we highlight the relevance of the two articles mentioned by the reviewer and include in the discussion a sentence about the concordance of the ecological niche model with the news reports of environmental isolations.

Lines at 475 "In the same way, this region was established by Mak S et al. 2015 as an optimal ecological niche for the isolation of the species C. neoformans [89]".

In this way and given the increase in publications associated with this topic, we consider that consolidating the work carried out by different researchers in this review article will make it possible to clarify the state of knowledge, highlight different reported niches, as well as identify regions, which, despite presenting similar climatic characteristics, report a low recovery of isolates. Likewise, we identified research needs in different regions and manage a consensus in the nomenclature previously used in the articles (following the recommendations given by the reviewers).

Point 2:  The format of the review: The results are presented as a series of summaries of the reviewed papers for each geographical area of the country, without an adequate analysis of the findings based on well-defined variables and using appropriate statistical tools. The methods should define the variables that will be taken in account for the analysis of data and, the analysis should be adequately performed by statistical tools for quantitative and qualitative parameters. On the contrary, you cannot reach solid conclusions.

Answer 2: To consolidate the findings reported by different groups, we conducted a systematic literature review, reporting what each author, based on the statistical methods implemented in the studies, found. Following the reviewers suggestions, we evaluated each article included in this review in terms of data analysis. We found that five of the 53 articles included in the review, had statistical analysis, the summarized statistical association with different variables including temperature, evaporation, precipitation, relative humidity, sunlight, soil pH, and sample type (excreta, debris, stem, fruits, leaves, soil, flowers), as detailed in results sections for each region, as following:

Line 212 – 214: For the statistical analysis, ANOVA, MANOVA, Canonical discriminant analysis, Pearson's correlation and Chi squared were performed, with the Statistical Analysis System (SAS) for Windows v. 8.02.

Line 250 – 252: The statistical analysis consisted of descriptive statistics and the Chi square test and the program used is not indicated.

Line 388 – 390: The statistical analysis consisted of the use of Non-parametric analysis, Tukey's hypothesis test and Chi squared, using the Statgraphics Centurion version 15.2 program.

Line 418 – 420: Descriptive statistics and the Chi-square test were used for statistical analysis using the SPSS Statistics 20.0 software under shareware license.

Line 438 – 439: One-way ANOVA, post hoc analysis, the Tukey test and the Chi-square using the test SPSS program version 15, for Statistical analysis.

To details the statistical correlation was included in the results and discussion section:

Lines 128 - 147: Of the included studies, five reports showed associations between isolates of Cryptococcus spp and environmental variables. Indeed, Granados et al. 2005 and Anacona et al. 2018 reported a correlation between high and mean temperature values and recovered isolates of C. neoformans [81], [18]. Conversely, C. gattii, serotype C, was fewer isolates from extreme minimum and maximum temperatures, according to Granados et al. 2005 [81]. Furthermore, a negative correlation with relative humidity was reported to C. neoformans, serotype A, and C. gattii serotype C [81], [23], in comparison with the favorable influence of humidity in recovered C. neoformans reported by Anacon 2018 et al [18]. Also, C. gattii, serotype B exhibited a positive correlation with relative humidity, associated with the high recovery under low evaporation conditions [81].

For the climatic variable precipitation, the C. neoformans serotype A evidenced a strong correlation [81]. For solar radiation, Granados et al. 2005 and Vallejo et al. 2016 reported that C. neoformans serotype A and C. gattii, serotype C exhibit a positive correlation with high solar radiation values contrast to C. gattii serotype B showed a positive correlation with low values of solar radiation [81], [23].

Regarding the sample source, Caicedo and Vallejo reported that the accumulation of excreta and the high presence of pigeons represent an increase in the recovery of the Cryptococcus neoformans fungus [28], [23]. 

Finally, Caicedo reported that wooden boxes and flat ceilings were the nests where there were more significant isolates of C. neoformans [28].

Line 488 – 495: Most studies do not perform data analysis except when reporting the percentages of positive samples found for each species according to the nomenclature used by each study. However, Granados et al. 2005 in Bogotá [81], Caicedo et al. 1996 with studies belonging to Cali [28], Quintero et al. 2005 of Cundinamarca [56], Contreras et al. 2018 [19], Vallejo et al. 2016 de Pasto [23]and Anacona et al. 2018 of Popayán [18], reported statistical data analysis. According to the results of these statistical analyzes, climatic variables that influence the recovery of the fungus are precipitation, temperature, evaporation, solar radiation, and humidity (Supplementary Table 1).

Point 3: The use of incorrect and outdated information, especially on the taxonomy of the genus and the current knowledge of the distribution of these yeasts worldwide and in different environments. The taxonomy of the genus has been revised and deeply modified in recent years based on new knowledge about its genotypes, phenotypes, ecological and clinical aspects. In fact, since 2017 there are two branches of researchers discussing the taxonomy within the two pathogenic Cryptococcus species complexes (as they are no longer single species). The two trends accept the existence of the C. neoformans and C. gattii species complexes with several different strains or species within the complex. As the differences between the yeasts constituting the two groups have much to do with their environmental distribution and pathogenicity, it is of notable importance to take this into account for the analysis intended in this manuscript. I recommend reading the following publications: 

Hagen F, Khayhan K, Theelen B, Kolecka A, Polacheck I, Sionov E, Falk R, Parnmen S, Lumbsch HT, Boekhout T. Recognition of seven species in the Cryptococcus gattii/Cryptococcus neoformans species complex. Fungal Genet Biol. 2015 May; 78:16-48. Epub 2015 Feb 23.

Hagen F, et al. Importance of Resolving Fungal Nomenclature: the Case of Multiple Pathogenic Species in the Cryptococcus Genus. mSphere. 2017 Jul-Aug; 2(4): e00238-17. doi: 10.1128/mSphere.00238-17

Kwon-Chung KJ, Bennett JE, Wickes BL, Meyer W, Cuomo CA, Wollenburg KR, Bicanic TA, Castañeda E, Chang YC, Chen J, et al. The Case for Adopting the "Species Complex" Nomenclature for the Etiologic Agents of Cryptococcosis. mSphere. 2017 Jan-Feb; 2(1). Epub 2017.

Answer 3: Based on the work published by Hagen et al. 2015 and Hagen et al. 2017 and Kwon-Chung et al. 2017, which provide highly relevant information, we include in the introduction, specifically in the taxonomy of the genus, updated information on the two branches reported in the literature for the classification of the Cryptococcus neoformans complex and the Cryptococcus gattii complex, as well as throughout the document, except for the information present in the results, since for this we keep the nomenclature used by the authors.

Lines 42 to 57: “C. neoformans complex has varieties (C. neoformans variety grubii serotype A, with three genotypes VNI, VNB, VNII and C. neoformans var. neoformans serotype D with genotype VNIV; the serotype hybrid AD is associated with the VNIII genotype) and the complex of species of C. gattii divided into four genotypes (VGI, VGII, VGIII, and VGIV) [6]–[9]. VGII has been subclassified into three associated genotypes: VGIIa, VGIIb, and VGIIc) [10], [11]. Nevertheless, the proposal to recognize the actual classification to C. neoformans species (known as C. neoformans variety grubii serotype A with VNB genotype, and VNI / VNII genotypes), C. deneoformans (referred to as C. neoformans var. neoformans with serotype D y genotype VNIV) and a hybrid composed of C. neoformans and C. deneoformans (with serotype AD and genotype VNIII). For the complex of species of C. gattii reorganized as five species: C. gattii with genotype VGI, C. deuterogattii with genotype VGII, C. bacillisporus with genotype VGIII, C. tetragattii with genotype VGIV, and finally C. decagattii with genotype VGIV and VGIIIc in addition to being hybrids such as C. deneoformans with C. gattii, the hybrid C. neoformans with C. gattii and the hybrid C. neoformans with C. deuterogattii [6], [7]. 

Lines 59, 61, 65, 72, 76, 79, 87, 93, 517, 546, 549, 553, 554, 556, 559, 560, 562: The word “complex” was added after the names C. neoformans and C. gattii in the introduction and discussion section.

Point 4:  Regarding the current knowledge of these yeasts in terms of genotypes, phenotypes, ecological and clinical aspects, since the major outbreak of C. deuterogattii cryptococcosis that occurred on Vancouver Island (British Columbia, Canada) more than 20 years ago (in 1999), many papers have been published demonstrating the presence of C. gattii complex species in temperate areas. Earlier ideas of tropical and subtropical distribution are now not as well supported. In addition, Eucalyptus is one of hundreds of tree species that have been shown to be ecological niches for different Cryptococcus species. In this regard, some reviews to update the information on pathogenic cryptococci and their ecology are very interesting. I recommend the following:

Robin C. May, Neil R.H. Stone, Darin L. Wiesner, Tihana Bicanic, Kirsten Nielsen. Cryptococcus: from environmental saprophyte to global pathogen. Nat Rev Microbiol. 2016; 14(2): 106–117. doi: 10.1038/nrmicro.2015.6

Carolina Firacative, Jairo Lizarazo, María Teresa Illnait-Zaragozí, Elizabeth Castañeda, Latin American Cryptococcal Study Group. The status of cryptococcosis in Latin America. Mem Inst Oswaldo Cruz. 2018; 113(7): e170554. doi: 10.1590/0074-02760170554

Massimo Cogliati, Roberta D'Amicis, Alberto Zani, Maria Teresa Montagna, Giuseppina Caggiano, Osvalda De Giglio, Environmental distribution of Cryptococcus neoformans and C. gattii around the Mediterranean basin. FEMS Yeast Res. 2016 Jun 1; 16(4): fow045. doi: 10.1093/femsyr/fow045

Jorge G. Carvajal, Alberto J. Alaniz, Mario A. Carvajal, Emily S. Acheson, Rodrigo Cruz, Pablo M. Vergara, Massimo Cogliati. Expansion of the Emerging Fungal Pathogen Cryptococcus bacillisporus Into America: Linking Phylogenetic Origin, Geographical Spread and Population Under Exposure Risk. Front Microbiol. 2020; doi:

Massimo Cogliati, Erika Puccianti, Maria Teresa Montagna, Antonella De Donno, Serdar Susever, Cagri Ergin, et al. Fundamental niche prediction of the yeast pathogens Cryptococcus neoformans and Cryptococcus gattii In Europe.  Environmental Microbiology 2017; 19: 4318-4325. DOI: 10.1111/1462-2920.1391510.3389/fmicb.2020.02117

Answer 4: The articles suggested by the reviewer were used to update the review manuscript.

The article by May et al. 2016 (Robin C. May, Neil RH Stone, Darin L. Wiesner, Tihana Bicanic, Kirsten Nielsen. Cryptococcus: from environmental saprophyte to global pathogen. Nat Rev Microbiol. 2016; 14 (2): 106-117. Doi: 10.1038 / nrmicro.2015.6) to clarify clinical aspects for Cryptococcus spp., as follows:

Line 58: "being C. neoformans the world cause of cryptococcal meningoencephalitis."

Line 63-64: "It should be noted that Cryptococcus spp is capable of remaining latent without affecting the host, which makes it highly adaptive."

The work of Li et al. 2020 (Y. Li, M. Zou, J. Yin, Z. Liu, and B. Lu, "Microbiological, epidemiological and clinical characteristics of patients with cryptococcal meningitis in a tertiary hospital in China: 6-year retrospective analysis, Front. Microbiol., Vol. 11, no. July, pp. 1-14, 2020), enriches ecological aspects, as follows:

Lines 76 - 78: "Also, C. neoformans complex was reported in tree species, with a higher prevalence compared to the C. gattii complex, for example, Eucalyptus spp [53], [54], Terminalia catappa [20] Olea europea [55] and Ceratonia (carob tree) [53].

Line 79: Conversely, the C. gattii complex has been described in temperate regions such as Canada, California, and Oregon [56].

The article by Cogliati et al. 2016 (Massimo Cogliati, Roberta D'Amicis, Alberto Zani, Maria Teresa Montagna, Giuseppina Caggiano, Osvalda De Giglio, Environmental distribution of Cryptococcus neoformans and C. gattii around the Mediterranean basin. FEMS Yeast Res. June 1, 2016; 16 (4): fow045. Doi: 10.1093 / femsyr / fow045), was included to update information on the primary and secondary niche of both complexes and climatic conditions, as follows:

Line 86 - 88: "Despite this, it is found in the environment to a lesser degree than the C. neoformans complex, especially those associated with the genus Eucalyptus [53]"

Line 502 - 503: "This maintains that bird droppings are a secondary and temporary niche of this species."

Line 508 - 509: “as reported by Cogliati et al. 2016 and 2017, who found a higher occurrence of the fungus in times of rain or wet periods".

Line 530 - 531: “and Cogliati et al. 2016 and 2017 found the associations with arboreal specimens as a niche and stable reservoir.

Line 535 - 537: "who establish the primary environmental source of C. neoformans as pigeon droppings" was replaced by "contrary to the findings reported by Cogliati et al. 2016 and 2017 that suggested that poultry feces is a secondary and transitory niche.

Line 541 - 543: "that is, at high temperatures, which contrasts with what was said by Cogliati et al. 2016 and 2017, who identified that the C. gattii complex does not tolerate low temperatures".

Reviewer 2 Report

This review describes the environmental distribution of the C. neoformans and C. gattii species complex in Columbia to determine specific environmental conditions where these fungi proliferate.

Comments:

  1. While this is a nice review of the literature of studies isolating Cryptococcus spp., since the aim was to determine the environmental conditions where these fungi grow, I think in addition to Table 1, adding another small table specifying the exact environmental conditions where C. neoformans and C. gattii have been isolated would be beneficial. For example, you can summarize the environmental conditions across studies where C. neoformans or C. gattii were isolated, including temperature, moisture, sunlight, sample type, etc. This would better describe places where we would now expect to isolate C. neoformans in the future.
  2. While overall the paper is fairly well written, there are a number of grammar mistakes. I suggest having the paper edited professionally or having a native English speaker edit the paper.

Minor comments:

  1. The term “department” is incorrectly used quite a bit in reference to different areas of Columbia. I suggest either deleting it entirely, for example on line 66, or replacing it with “region”.
  2. The sentence on lines 106-107 can be deleted.
  3. The phrase, “In the same way” is used quite a bit at the beginning of a sentence. In most cases, it can be deleted.
  4. When discussing a figure in the text, “figure” should be capitalized (…in Figure 2).
  5. The phrase “On the other hand” is almost never used in scientific writing unless it is preceded with “On the one hand” in a previous sentence.  In most cases, you can replace this phrase with “In contrast” or “However”.
  6. There are a number of places in the text where reference numbers do not have brackets around them (lines 255, 291, and 532).
  7. There are a number of genus and species names that should be italicized (lines 72, 145, 334, and 363-364).
  8. Please add a reference for the sentence on lines 541-543.

Author Response

Response to Reviewer 2 Comments

This review describes the environmental distribution of the C. neoformans and C. gattii species complex in Columbia to determine specific environmental conditions where these fungi proliferate.

Comments:

Point 1: While this is a nice review of the literature of studies isolating Cryptococcus spp., Since the aim was to determine the environmental conditions where these fungi grow, I think in addition to Table 1, adding another small table specifying the exact environmental conditions where C. neoformans and C. gattii have been isolated would be beneficial. For example, you can summarize the environmental conditions across studies where C. neoformans or C. gattii were isolated, including temperature, moisture, sunlight, sample type, etc. This would better describe places where we would now expect to isolate C. neoformans in the future.

Answer 1: Thank you very much for your comments, it helps us to improve the information in our writing, we have made complementary table 1 (Supplementary Table 1), where we expose the most specific of environmental conditions which favor the development of the fungus.

Point 2: While overall the paper is fairly well written, there are a number of grammar mistakes. I suggest having the paper edited professionally or having a native English speaker edit the paper.

Answer 2: Thank you for your comments on the language. A specialized English speaker is actually reviewing the manuscript. We hope to have the language fully revised before publication.

Minor comments:

Point 1: The term "department" is incorrectly used quite a bit in reference to different areas of Columbia. I suggest either deleting it entirely, for example on line 66, or replacing it with "region".

Answer 1: The correction was made throughout the document, replacing the words department by region.

Point 2: The sentence on lines 106-107 can be deleted.

Answer 2: The phrase "Environmental isolates of the Cryptococcus neoformans species complex and the Cryptococcus gattii species complex in Colombia" was deleted.

Point 3: The phrase, "In the same way" is used quite a bit at the beginning of a sentence. In most cases, it can be deleted.

Answer 3: The term in line 152 was eliminated, substituting it for "Reporting other isolations." On line 459, it was changed to "Demonstrating that," and on line 584, it was modified to "Similarly."

Point 4: When discussing a figure in the text, "figure" should be capitalized (… in Figure 2).

Answer 4: Following the recommendation throughout the text, the word figure was corrected by "Figure"

Point 5: The phrase "On the other hand" is almost never used in scientific writing unless it is preceded with "On the one hand" in a previous sentence. In most cases, you can replace this phrase with "In contrast" or "However".

Answer 5: following the recommendation, the phrase "On the other hand" was replaced throughout the text. Lines 60, 79, 220, 259, 351, 573 and 584.

Point 6. There are a number of places in the text where reference numbers do not have brackets around them (lines 255, 291, and 532).

Answer 6:  Bracket correction was made on the text.

Point 7: There are a number of genus and species names that should be italicized (lines 72, 145, 334, and 363-364)

Answer 7: The proper correction of genus and species names was made by changing them to italics (lines 88, 179, 356 and 386 - 387).

Point 8: Please add a reference for the sentence on lines 541-543. It was not added.

Answer 8: The reference "[88]" (S. Mak, N. Vélez, E. Castañeda, P. Escandón, and CES Group, "The Fungus among Us: Cryptococcus neoformans and Cryptococcus gattii Ecological Modeling for Colombia," J. Fungi, vol. 1, no. 3, pp. 332–344, 2015.)

Reviewer 3 Report

The authors have comprehensively reviewed Cryptococcus ecology in Columbia based on environmental sampling efforts. The article--tables/figures in particular--are well-organized. Moderate English language editing is likely needed. 

The authors may want to consider considerably reducing the length of the “Results” section.

Table 1 already provides a much more readable summary of the sampling performed in different regions of Columbia. The authors could cut redundant information from the “Results” section so that their discussion is more prominent.

Author Response

Response to Reviewer 3 Comments

Point 1: The authors have comprehensively reviewed Cryptococcus ecology in Columbia based on environmental sampling efforts. The article--tables/figures in particular--are well-organized. Moderate English language editing is likely needed. 

Answer 1: Thank you for your recognition. A specialized English speaker is actually reviewing the manuscript. We hope to have the language fully revised before publication.

Point 2: The authors may want to consider considerably reducing the length of the “Results” section. Table 1 already provides a much more readable summary of the sampling performed in different regions of Columbia. The authors could cut redundant information from the “Results” section so that their discussion is more prominent.

Answer 2:  Thank you very much for your comments, we have eliminated redundant information from each study in the results section, in the same way modifying some lines to reduce it as follows:

Line 312: The paragraph "Ten public areas were collected during October 2016 (Mercedes Abrego, Simón Bolívar, Arcoíris (La Libertad) and the area around the General Santander Stadium), January 2017 (Antonia Santos Park, La Victoria, Juana Rangel de Cuellar, and Santander Park) and April 2017 (National Parks and Fuentes de Leones)" was reduced to "in ten public areas"

Line 417: The paragraph "The number of isolates was lower in fresh samples; however, it was evident that those that were wet (not necessarily fresh) provided a higher number of positive isolates, similar to that reported by Granados and Castañeda 2006 [87]" was reduced to "The number of isolates was lower in fresh samples, and wet samples (not necessarily fresh) provided a higher number of positive isolates, similar to that reported by Granados and Castañeda in 2006 [90]."